# Self-consistent Global Transport of Metallic Ions with WACCM-X

Jianfei Wu[1,2,3], Wuhu Feng[4,5], Han-Li Liu[6], Xianghui Xue[1,2,3,7,8], Daniel Robert Marsh[6,9], and John Maurice Campbell Plane[4]

[1]CAS Key Laboratory of Geospace Environment, School of Earth and Space Sciences, University of Science and Technology of China, Hefei, China
[2]CAS Center for Excellence in Comparative Planetology, China
[3]Mengcheng National Geophysical Observatory, School of Earth and Space Sciences, University of Science and Technology of China, Hefei, China
[4]School of Chemistry, University of Leeds, Leeds, UK
[5]National Center for Atmospheric Science, University of Leeds, Leeds, UK
[6]National Center for Atmospheric Research, Boulder, Colorado, USA
[7]Hefei National Laboratory for the Physical Sciences at the Microscale, University of Science and Technology of China, Hefei, China
[8]Frontiers Science Center for Planetary Exploration and Emerging Technologies, University of Science and Technology of China, Hefei, China
[9]Faculty of Engineering and Physical Sciences, University of Leeds, Leeds, UK

**Correspondence:** Xianghui Xue (xuexh@ustc.edu.cn) and John M. C. Plane (J.M.C.Plane@leeds.ac.uk)

**Abstract.** The NCAR Whole Atmosphere Community Climate Model with thermosphere and ionosphere eXtension (WACCM-X) v2.1 has been extended to include the neutral and ion-molecule chemistry and dynamics of three metals (Mg, Na, and Fe), which are injected into the upper mesosphere/lower thermosphere by meteoric ablation. Here we focus on the self-consistent electrodynamical transport of metallic ions in both the E and F regions. The model with full ion transport significantly improves the simulation of global distribution and seasonal variations of $Mg^+$, although the peak density is slightly lower (about 35% lower in peak density) compared with the SCIAMACHY measurements. Near the magnetic equator, the diurnal variation in upward and downward transport of $Mg^+$ is generally consistent with the "ionosphere fountain effect". The thermospheric distribution of Fe is shown to be closely coupled to the transport of $Fe^+$. The effect of ion mass on ion transport is also examined: the lighter ions ($Mg^+$ and $Na^+$) are transported above 150 km more easily than the heavy $Fe^+$. We also examine the impact of the transport of major molecular ions, $NO^+$ and $O_2^+$, on the distribution of metallic ions.

## 1   Introduction

The presence of layers of meteor-ablated metal atoms between 80 and 105 km has been known for decades (Plane et al., 2015). More recently, there have been a growing number of observations of the thermospheric metal layers up to 200 km. For example: Chu et al. (2011) reported neutral Fe layer observations up to 155 km at McMurdo, Antarctica (77.8°S, 166.7°E); Gao et al. (2015) found that several observations of Na layers reached up to 170 km, using a large-aperture astronomical telescope at Lijiang, China (26.7°N, 100.0°E); and Friedman et al. (2013) investigated a descending thermospheric K layer from altitudes above 155 km at Arecibo, Puerto Rico (18.35°N, 66.75°W). These high altitude thermospheric neutral metal

layers are challenging to explain, since ablation occurs predominantly below these altitudes, which raises interesting questions regarding their formation mechanisms. Neutral metal atoms and their corresponding atomic ions are tightly coupled through ionization (via photo-ionization or charge transfer with ambient ions e.g. $NO^+$ and $O_2^+$), and neutralization (via dielectronic recombination with electrons, or dissociative electron recombination if they have formed a molecular ion) (Plane et al., 2015). Thus, the vertical and horizontal transport of metallic ions is a key process for controlling the behavior of neutral atoms in the thermosphere (E and F regions). Interestingly, the reported occurrence of thermospheric metal layers appears to show great geographical variability, underlining the importance of global ion transport. In addition, metallic ions play a central role in the formation of thin, concentrated layers of ions in the E-region (sporadic E layers, or Es), which affect radio transmission (e.g., Narcisi, 1968; Layzer, 1972; Plane et al., 2015; Yu et al., 2021).

Thus far, a number of modeling studies have attempted to simulate the transport of metal ions in the thermosphere and, more recently, the role of metallic ions in the formation of the thermospheric metal layers. For example, Carter and Forbes (1999) developed a two-dimensional (2-D) model to examine both global and local transport of $Fe^+$ ions; Chu and Yu (2017) used a thermosphere-ionosphere $Fe/Fe^+$ model to investigate the formation of thermospheric Fe layers observed by lidar at McMurdo, Antarctica; and Cai et al. (2019) conducted a 2-D simulation of $Na/Na^+$ in the E and F regions to explore the formation of thermospheric sodium layers observed at the Andes Lidar Observatory (30.25°S, 70.74°W). Recently, Huba et al. (2019) have included the global transport of metallic ions ($Mg^+$ and $Fe^+$) into the SAMI3 model of the E and F regions, and concluded from a two-day model run that ions are redistributed by the combined effects of the neutral wind and electric field. However, no previous studies appear to have examined the full transport of metal ions in a self-consistent global chemical-dynamical model, incorporating the full life cycle of the thermospheric metal atom and metal ion chemistry and the injection of these metals from meteoric ablation.

This paper describes the development of a new global model of three metal species (magnesium, iron, and sodium) in the E and F regions, to gain a detailed understanding of the effects of multi-scale atmospheric motions from the lower atmosphere and solar activity on the thermospheric metal layers. The main contribution reported here is to incorporate a self-consistent solution of full global transport of $Mg^+$, $Fe^+$ and $Na^+$ in both the E and F regions, within the chemistry-climate WACCM-X 2.1 model, in addition to a detailed description of the neutral and ion-molecule chemistry of these metals and the meteoric ablation source required to model the metal atom layers around 90 km. This is described in Section 2. Section 3 presents the findings of the model simulation, focusing on the seasonal and the diurnal variation of ions and the effects of ion electro-dynamical transport. The final section includes a brief summary and a discussion of future directions with the model.

## 2  Model Description and Ion Transport

### 2.1  Model Description

WACCM-X is extended in the present study to include the full life cycle of meteoric metals combined with interactive chemistry, dynamics, deposition and ion transport in the ionosphere. WACCM-X is an atmospheric component of the Community Earth System Model (CESM, version 2.1.3; Hurrell et al., 2013) developed by the National Center for Atmospheric Research.

The key chemistry and dynamical features are based on CAM4 and WACCM4 and are described in detail in Marsh et al. (2013b), and Neale et al. (2013). Validated metal chemistry modules for magnesium (Langowski et al., 2015), sodium (Marsh et al., 2013a), and iron (Feng et al., 2013) with updated rate coefficients from Plane et al. (2015), Bones et al. (2016) and Viehl et al. (2016), are added. The meteoric input functions (MIFs) were estimated from the Leeds Chemical ABLation MODel (CABMOD-3) combining with an astronomical model (Carrillo-Sánchez et al., 2020). The transport of the neutral and ionized metallic species by eddy/molecular diffusion and winds is treated in the same way as the transport of most active chemical species (for example, $O_3, CO_2$ etc.).

A detailed description of WACCM-X 2.0 is provided by Liu et al. (2018a) and a brief summary is given here. The model has some key features and improvements since WACCM-X 1.0 (Liu et al., 2010), including a self-consistent electrodynamics module, F-region $O^+$ transport, a solver for electron and ion temperatures, and reduction in the damping coefficient of atmospheric tides (Liu et al., 2018b). The model top is set at $4.1 \times 10^{-10}$ hPa ($\sim$500 to $\sim$700 km, depending on solar activity), with a vertical resolution of a quarter of a scale-height in the mesosphere and thermosphere. The horizontal resolution is $1.9°$ in latitude and $2.5°$ in longitude, and all model results used in the paper are from one-year free-running simulations perpetual year 2000AD under solar medium conditions (constant F10.7=130 sfu (solar flux units) and Kp=1) with an output frequency of 1 hr. The $O^+$ transport method is described in detail by Liu et al. (2018a).

## 2.2 Ion Transport Equation

Since meteoric ablation, deposition, transport by the neutral winds, eddy and molecular diffusion, chemical production, and loss are already contained in the metal chemistry modules, the metal ion transport is calculated separately, in a similar way to the treatment of $O^+$ transport described by Liu et al. (2018a). The continuity equation of metal ion transport can be simplified as:

$$\frac{\partial n_i}{\partial t} = -\nabla \cdot (n_i \boldsymbol{V}_i) \tag{1}$$

where $n_i$ represents the number density of metal ions, and $\boldsymbol{V}_i$ is the ion transport velocity. The ion transport velocity is adapted from the derivation described by Carter and Forbes (1999) and Chu and Yu (2017), extended to a 3-D global model:

$$\boldsymbol{V}_i = \frac{\xi}{1+\xi^2} \frac{\boldsymbol{V}_n \times \boldsymbol{B}}{B} + \frac{1}{1+\xi^2} \frac{(\boldsymbol{V}_n \cdot \boldsymbol{B}) \boldsymbol{B}}{B^2}$$
$$+ \frac{\xi}{1+\xi^2} \frac{\boldsymbol{E}}{B} + \frac{1}{1+\xi^2} \frac{\boldsymbol{E} \times \boldsymbol{B}}{B^2} + \boldsymbol{V}_{ambi} \tag{2}$$

where $\boldsymbol{E}$ and $\boldsymbol{B}$ are the electric field and the Earth's magnetic field, respectively. $\boldsymbol{V}_n$ is the neutral wind, and $\xi = \frac{\nu_{in}}{\omega_i}$ is the ratio of ion-neutral collision frequency (in the laboratory frame-of-reference (Banks and Kockarts (1973)) to the ion gyrofrequency. $\boldsymbol{V}_{ambi}$ is the ion velocity due to ambipolar diffusion, derived from Schunk and Nagy (2000, equations (5.54) and (5.70)), where we treat the metallic ions as minor ions and the ion with higher density in $NO^+$ and $O^+$ as a major ion species. Note that the term corresponding to transport is omitted from Equation 2, because advection by the neutral wind is already included in the dynamical core of WACCM-X. The first two terms are related to contributions by the neutral winds, with the first term being the V×B drift i.e. the Lorenz force. The third and fourth terms are due to the electric field. A Flux-Corrected

Transport (FCT) algorithm (Boris et al., 1993) is applied to compute the vertical transport velocity; this algorithm is designed for solving steep density gradients.

As mentioned above, charge transfer from molecular ions such as $NO^+$ and $O_2^+$ to metal atoms provides a major source of metallic ions. However, due to the short lifetime of these two molecular ions ($\sim$5 mins in the daytime), they were assumed to be in chemical equilibrium in WACCM-X 2.0. In this study, the transport of molecular ions $NO^+$ and $O_2^+$ is now considered along with the metal ions. Simulation results with and without the molecular ion transport are compared to determine the impact on the metal ion distribution.

## 90 3 Results and Discussion

$Na^+$, $Fe^+$, and $Mg^+$ undergo similar transport forces in the E and F regions, apart from the effect of their mass differences. Because $Mg^+$ is the only one of these ions for which near-global observations are available (Langowski et al., 2015), we focus here on the results for $Mg^+$. However, we also explore $Fe/Fe^+$ ion-neutral coupling in the formation of thermospheric metal layers, since Fe is the most abundance and widely studied thermospheric metal species. In addition, the $Mg^+/Fe^+$ and 95 $Mg^+/Na^+$ ratios are employed to examine the effect of metal ion mass difference, because $Fe^+$ is more than twice as heavy as $Mg^+$ (56 versus 24 amu), while the masses of $Na^+$ and $Mg^+$ are very similar (23 versus 24 amu).

### 3.1 Seasonal variation of $Mg^+$ simulated by WACCM-X

Figure 1 shows the monthly mean density of $Mg^+$ as a function of latitude and altitude, where the monthly mean data is zonally averaged. An obvious seasonal signal is exhibited with clear latitudinal dependence, which is generally consistent with 100 the SCIAMACHY measurements (Figure 6 in Langowski et al. (2015)). Nevertheless, WACCM-X $Mg^+$ density is slightly underestimated relative to SCIAMACHY observations (by $\sim$ 35%), which is likely related to the MIF used in the simulation. The peak altitude of $Mg^+$ in the summer hemisphere at middle latitudes ($\sim$ 40°$\pm$10°) is $\sim$10 km higher than at other latitudes, in accord with observations (Langowski et al., 2015). This appears to be caused by the vertical ion velocity due to the neutral wind (first and second terms in Eq. 2), with the monthly mean drift velocities shown in Figure 1. Since the metallic ions are 105 the main reservoir for neutral metal atoms in the lower thermosphere (Plane et al., 2015), this is in good agreement with the summer peak occurrence of lower thermospheric neutral Na layers observed by mid-latitude lidars (Wang et al., 2012; Dou et al., 2013; Yuan et al., 2014; Xun et al., 2020).

In contrast to the SCIAMACHY measurements, which show a minimum at the equator, the WACCM-X simulation shows a maximum in peak altitude and number density at the equator. Note that the SCIAMACHY observations are made at a particular 110 local time of around 10:00 LT, whereas the WACCM-X data in Figure 1 is a diurnal and zonal average. To address this, we also present the simulation results at the same local time (10:00 LT) in Figure 2, and they are in better agreement with SCIAMACHY observations. Another noteworthy feature is that this simulation shows the pronounced maximum in peak altitude and density at $\sim$45° (N/S) at 10 LT, in accord with the SCIAMACHY observations, which is absent in WACCM-Mg in Langowski et al. (2015). However, the number densities at 10 LT are even lower, and the peak altitude is about 5 km lower than the daily

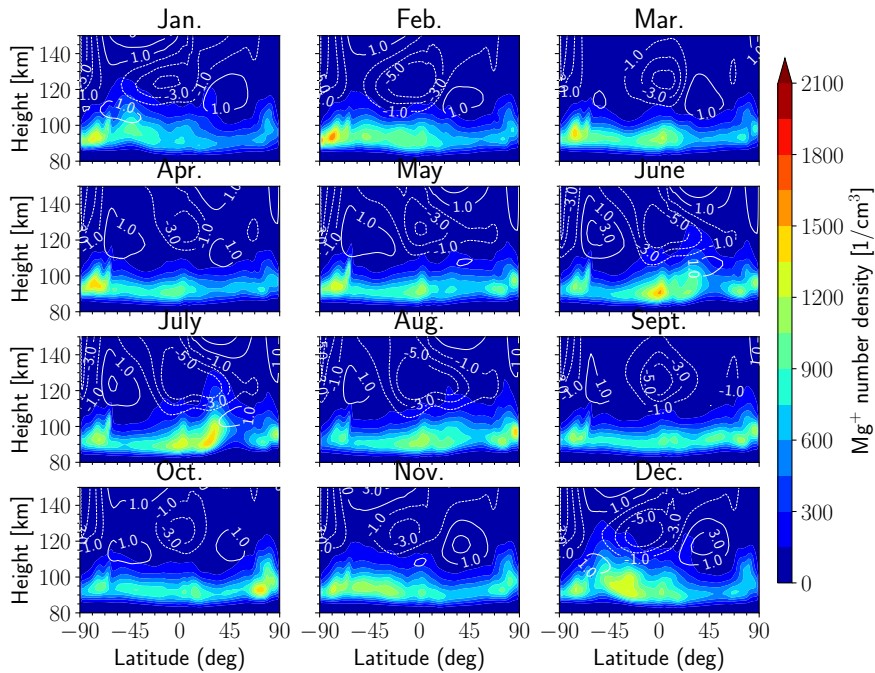

**Figure 1.** Monthly and zonal mean of the $Mg^+$ number density (color contours) and the drift velocity (in m/s) due to the neutral wind (line contours) as a function of latitude and altitude in different months.

averaged simulation ( Figure 1). The pronounced discrepancy between Figure 2 and SCIAMACHY measurements is probably due to the diurnal variation of ion electro-dynamical transport, which should be further investigated in the future.

Figure 3 (a-d) presents the global geographical distribution of $Mg^+$ column densities at the equinoxes and solstices, and at 00 UTC. The white dashed line depicts the dip equator. Despite the high accumulation of column density at the poles, the $Mg^+$ column density exhibits relatively high values at lower and middle latitudes in the summer hemisphere (Figure 3 (b) and (d)). This is in good agreement with the conclusion of a summertime maximum shown in Figure 1. Interestingly, the geographic distributions in all four seasons show stronger convergences along the magnetic equator, which might be related to the modulation of the equatorial fountain effect.

In Figure 3 (e) we show the seasonal variation of zonal-averaged $Mg^+$ column density at different latitudes. The modelled $Mg^+$ column exhibits a maximum in the summer hemisphere at middle and high latitudes, which is in general agreement with the results derived from the SCIAMACHY measurements and WACCM-Mg model results (Figure 9 and Figure 16 in Langowski et al. (2015)), and the SBUV nadir measurements (Figure 9 in Joiner and Aikin (1996)). The partial column density above 110 km still shows a similar seasonal variation, which is probably due to thermospheric ion transport (not shown).

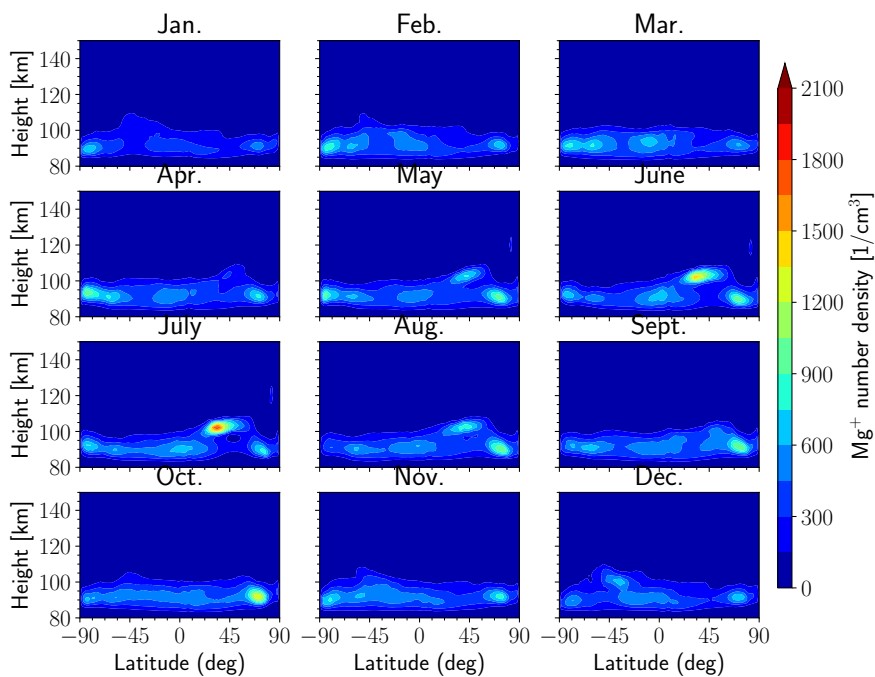

**Figure 2.** Monthly mean $Mg^+$ number density obtained by averaging data at 10 LT for all longitudes as a function of latitude and altitude in different months.

## 3.2 Diurnal variations of $Mg^+$ simulated by WACCM-X

To investigate the diurnal variation of metallic ions in the model, we present the $Mg^+$ number density (on a log scale) as a function of latitude and altitude at the December Solstice and $0°$ longitude; the panels show universal times (i.e. local times) of 00, 06, 12 and 18 UT (Figure 4 (a-d)). The white dotted line denotes the $F_2$-layer height of the peak electron density ($hmF_2$). This shows that the strongest diurnal variations are found in equatorial and high latitudes. Here we focus on the "fountain effect" on ion transport, where the equatorial ions are first lofted to higher altitudes via E×B motion, and then drift down along the magnetic field lines (Kelley, 2009). $Mg^+$ is expected to be lofted to high altitudes ($\sim 400$ km) by the E×B drift above the magnetic equator during the day, because of the daytime eastward electric fields (Huba et al., 2019). Figure 4 (e) shows the diurnal variation of $Mg^+$ number density near the dip equator ($12°N, 0°$) as a function of local time; the upward drift of $Mg^+$ peaks at around 20 LT, and the $Mg^+$ then drifts down towards the main layer at midnight ($\sim 02$ LT). Note that the reduction of $Mg^+$ at the equator at 06 LT confirms that the ion density at the equator is largely dependent on local time (Figure 4 (b) and (e)). The phase of $Mg^+$ diurnal variation shows a high correlation with variations in the electron density (the change of $hmF_2$). Instead of being redistributed along the magnetic field lines to the subtropical region by the "classical" fountain effect (e.g.,

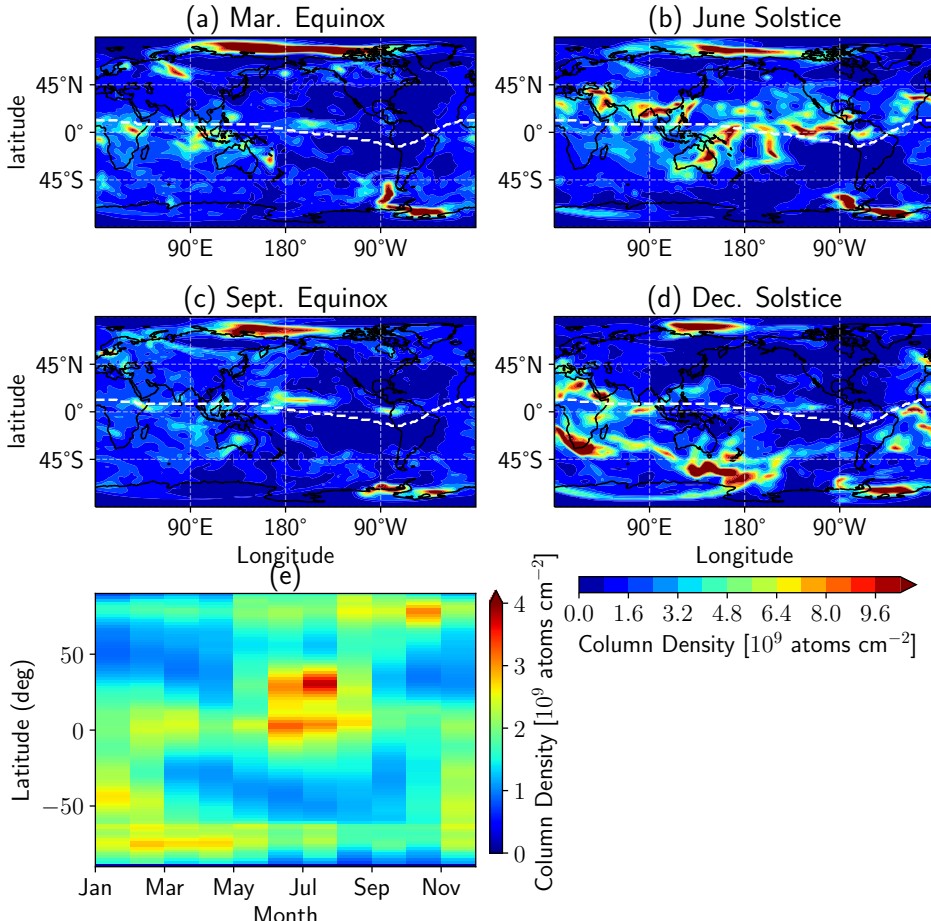

**Figure 3.** (a-d) Global geographical distribution of the Mg$^+$ column density during the equinoxes and solstices, at 00 UTC. (e) Seasonal variation of the zonal mean Mg$^+$ column density as a function of month and latitude. The white dashed lines indicates the position of the dip equator.

Pi et al., 2009), the $Mg^+$ shows a more complex downward trajectory, i.e., the ions are not transported symmetrically to both sides of the geomagnetic equator, which is closer to the scenario proposed by Cai et al. (2019).

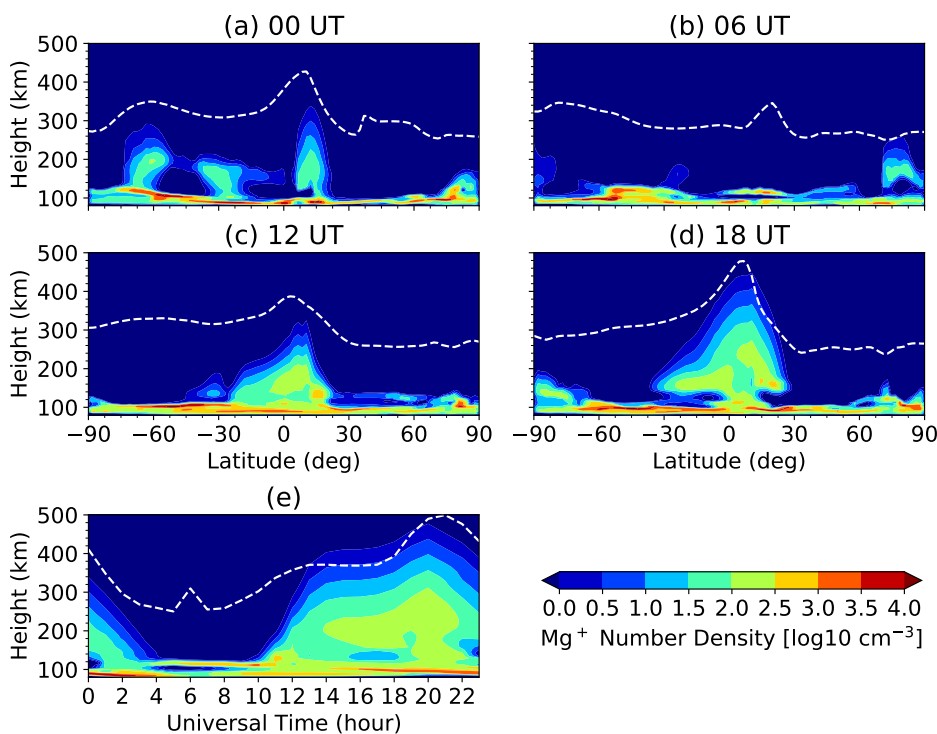

**Figure 4.** (a-d) $Mg^+$ density as a function of latitude and altitude at the December Solstice and longitude $0°$, at four universal times (local times): 00, 06, 12, 18 UT. (e) $Mg^+$ density over ($12°N$, $0°$) as a function of universal time and altitude. The white dashed lines denote $hmF_2$.

### 3.3 Fe and $Fe^+$ vertical profile comparison

In order to demonstrate the effect of electro-dynamical transport on the metals, a standard simulation without metal ion transport

was performed (termed the control run). Figure 5 shows the $Fe^+$ and Fe vertical profiles for the ion transport run (solid lines), and control run (dotted/dashed lines) at the equinoxes and solstices. Three geographic latitudes at a longitude of $180°$ are chosen for comparison: $0°$ for the magnetic equator, $20°S$ for the subtropical region corresponding to the fountain effect downward drift, and $45°S$ for the middle latitude corresponding to the summertime peak altitude. Without the ion transport (the control run), both Fe and $Fe^+$ exhibit roughly Gaussian-shaped layers with peak heights between 90 and 100 km (dotted/dashed lines).

When ion transport is turned on, the vertical profiles of $Fe^+$ vary depending on latitude and season. For instance, $Fe^+$ near the equator is always transported to a higher altitude (blue lines), consistent with the fountain effect in the dip equator region. However, ions at subtropical ($20°S$) (orange lines) and middle ($45°S$) latitudes (green lines) exhibit quite different transport motions. At the December solstice at midnight, $Fe^+$ is transported to a high level ($\sim1$ $cm^{-3}$ at 200-300 km) at middle and

subtropical latitudes, which is related to the dominant upward drift velocity due to neutral wind (not shown). In other seasons, these ions can be transported both upward and downward, depending on the vertical ion velocity.

Since the neutral atoms are not directly transported by the electromagnetic field, they are influenced by ions through the recombination between ions and electrons. The last two panels in Figure 5 illustrate the Fe atom distributions. In general, changes in Fe (at densities $> 1$ cm$^{-3}$) do not appear to be synchronized with the upward transport of Fe$^+$, so that the vertical distribution of Fe in the transport run is similar to that in the control run. However, there is an obvious increase in high-altitude Fe (i.e. above $\sim$140 km) in the equatorial region (blue line), corresponding to the upward transport of ions; and at the December solstice around midnight, the Fe number density above 150 km is much higher than that in the control run at all southern latitudes.

## 3.4 Effect of metal ion mass on transport

Figure 6 compares the Mg$^+$/Fe$^+$ ratio (left panel) and the Mg$^+$/Na$^+$ ratio (right panel) as a function of height and latitude at the equinoxes and solstices. Note that changes in the ratios below 100 km are due to differences in the ion-molecule chemistries of the metals (Plane et al., 2015), which is not the focus here. There are several advantages in choosing these three metallic ions. First, Fe$^+$ is more than twice as heavy as Mg$^+$. Second, Fe$^+$ is much heavier, and Mg$^+$ is slightly lighter, than the mean mass of air molecules in the E region. Third, Na$^+$ has a comparable mass with Mg$^+$. As expected, the lighter ions are transported above 150 km more easily than the heavy Fe$^+$, so that the Mg$^+$/Fe$^+$ ratio increases from $\sim$1 at 120 km to $>$2 above 150 km and to $>$ 30 above 300 km. By the same token, the Mg$^+$/Na$^+$ ratio shows very little change above 120 km.

The zonally-averaged Fe$^+$/Mg$^+$ ratio below 200 km simulated by WACCM-X also accords with the limited available observations (Dymond et al., 2003; Kumar and Hanson, 1980), which showed that the average Fe$^+$/Mg$^+$ ratio is around 1.5:1. The present study also simulates the extreme variability of the Fe$^+$/Mg$^+$ ratio above 300 km (as low as 1:50 in Kumar and Hanson (1980)). However, the unexpectedly large Fe$^+$/Mg$^+$ ratio ($\sim$10–50) reported by Dymond et al. (2003) is only captured at a few points about 150 km in our model (not shown). Interestingly, the striking differences between distinct thermosphere-ionosphere Fe and diffuse thermosphere-ionosphere Na reported by Chu et al. (2020) are thought to be related to mass separation. There is no question that more observations are needed to confirm and validate these findings.

Figure 6 shows that whereas the Mg$^+$/Fe$^+$ ratios (left panel) show relatively little latitudinal variation compared with the variation with altitude (mostly caused by the mass difference of the ions), the Mg$^+$/Na$^+$ ratios (right panel) show marked interhemispheric differences at the solstice periods. The reason for this is the difference in the ion-molecule chemistry of Na$^+$, compared with Mg$^+$ (and Fe$^+$). The Na$^+$ ion has a closed electronic shell (it is isoelectronic with the inert gas Ne), and so does not react with O$_3$, in contrast to Mg$^+$ and Fe$^+$ (Plane et al., 2015). Formation of MgO$^+$ by the fast reaction with O$_3$ is the main route to neutralization of Mg$^+$ above 90 km (Whalley et al., 2011). During summer at mid- to high-latitudes ($> 30°$), O$_3$ above 90 km is heavily depleted through a combination of longer diurnal photolysis and reaction with the elevated levels of H produced from H$_2$O which upwells over the summer pole (Plane et al., 2015). In contrast, the O$_3$ density in the lower thermosphere is more than an order of magnitude higher in the winter polar vortex. The result is that lower thermospheric Mg$^+$ ions at latitudes higher than $\sim$30° are relatively long-lived in summer, and can be transported vertically throughout the

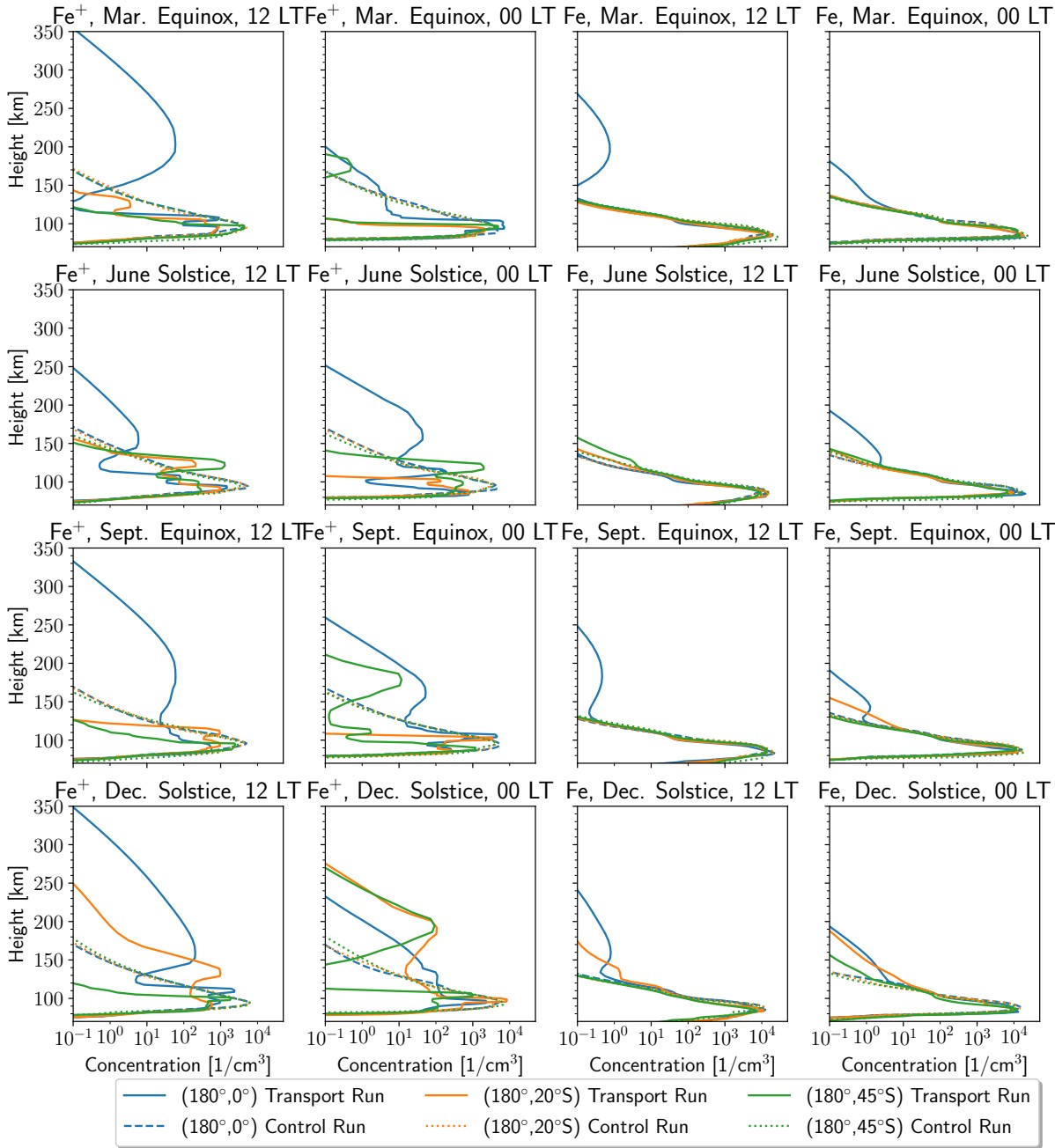

**Figure 5.** Comparison of the vertical profiles of the $Fe^+$ (left two panels) and Fe (right two panels) number densities, for the control run (dotted/dashed lines) and run with ion transport (solid lines) over selected geographic latitudes, at the equinoxes and solstices and midday (12 LT) and midnight (00 LT).

thermosphere. This leads to the higher $Mg^+/Na^+$ ratios in the summer hemisphere, as shown in Figure 6. The converse operates at latitude higher than $30°$ in the winter hemisphere, where the relatively high $O_3$ tends to neutralize lower thermospheric $Mg^+$.

Because $Na^+$ and $Mg^+$ have very similar masses, their ratios above 100 km are fairly constant with height.

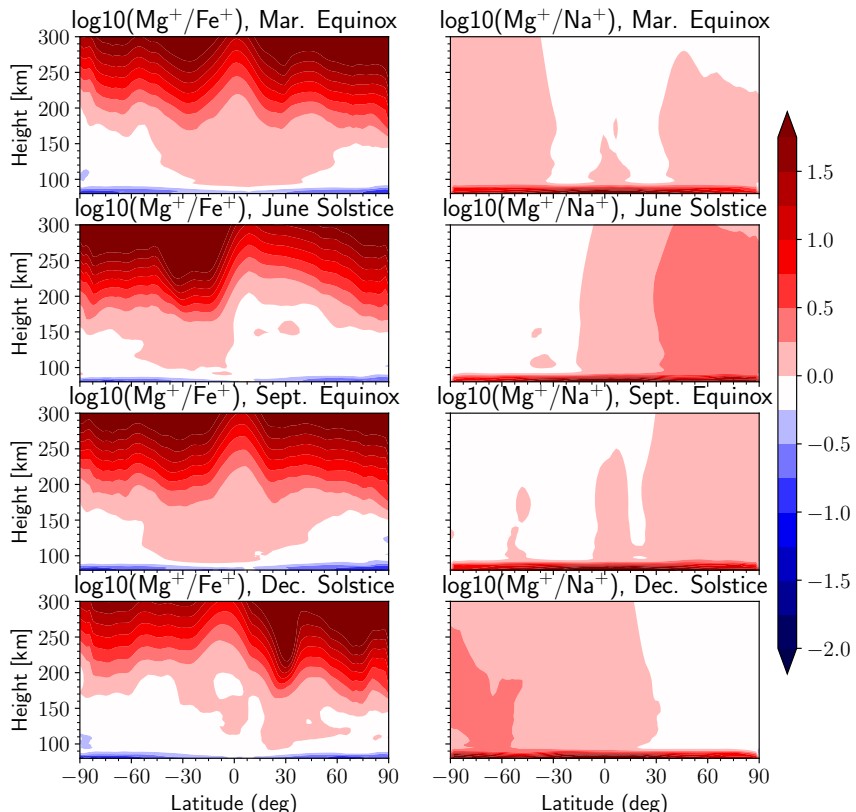

**Figure 6.** The zonally-averaged $Mg^+/Fe^+$ ratio (left panel) and $Mg^+/Na^+$ ratio (right panel) versus height and latitude at the equinoxes and solstices.

## 3.5  Effect of $NO^+$ and $O_2^+$ transport

Figure 7 and 8 compare the distribution of metal ions from two simulations, one with the transport of all ions (hereafter TA), and the other with $NO^+$ and $O_2^+$ in chemical equilibrium (hereafter CE). Note that the simulation results presented herein investigate the potential impacts of the transport of two major molecular ions on the distribution of metallic ions and do not

provide a comprehensive review of all possible effects.

Figure 7 shows the monthly mean density of $Mg^+$ as a function of latitude and altitude in June for the TA (Figures 7a) and CE (Figures 7b) simulations, along with their difference (Figures 7c), where the monthly mean data is zonally averaged. In general, there is a good correspondence between the two simulations in terms of the latitude-altitudinal distribution of the monthly mean $Mg^+$ density. As seen in Figure 7c, the peak density in CE simulation is generally a little higher than that in the TA simulation, especially in the high latitudes of the southern hemisphere. Figure 8 compares the diurnal variation of $Mg^+$ in the two simulations. Both cases simulate the significant "fountain effect", which was discussed in Section 3.2. With the transport of major molecular ions, the peak height of the metal layer after midnight is higher. As discussed by Plane et al. (2015), the charge transfer of neutral metal atoms with $NO^+$ and $O_2^+$ is the major sources of metallic ions in the E region. Due to the very short lifetimes of $NO^+$ and $O_2^+$ during daytime, the transport of these molecular ions between model grid-boxes has little effect on the metallic ions. In contrast, the reduced densities of the molecular ions (and electrons) at night means that their increased lifetimes become comparable to transport time-step. Additional metallic ions are therefore produced via charge transfer with the downward transport of $NO^+$ and $O_2^+$ at night in the TA simulation.

## 4  Conclusions

The WACCM-X high altitude chemistry-climate model has been extended to incorporate the full life cycle of multiple meteoric metal ions and atoms (Mg, Na, and Fe, currently). A major advantage of WACCM-X is the self-consistent treatment of dynamics and electrodynamics allowing us to quantitatively investigate the global distribution of metal ions and the formation mechanisms of thermospheric metal layers. The present study explores, for the first time, the seasonal variations of thermospheric metal ions by including global metal ion transport in the E and F regions.

There are a number of interesting findings: (1) A clear seasonal cycle is found in the monthly averaged global distributions of $Mg^+$, in good agreement with the SCIAMACHY measurements (Langowski et al., 2015), although the peak height and peak density are about 5 km and 35% lower than the observations, respectively. (2) Uplift of metal ions in the summer hemisphere at mid-latitudes ($\sim 40° \pm 10°$), driven by the vertical ion velocity due to the neutral wind, appears to explain the summer peak occurrence of lower thermospheric neutral Na layers observed by mid-latitude lidars (Wang et al., 2012; Dou et al., 2013; Yuan et al., 2014; Xun et al., 2020). (3) Upward transport of metallic ions by E×B forcing is generally consistent with the "fountain effect". (4) The formation of thermospheric neutral metal layers is strongly influenced by the upward transport of ions, since metallic atoms and ions are coupled by relatively fast reactions in the lower thermosphere (Plane et al., 2015). (5) A pronounced mass separation of $Fe^+$ with the two lighter ions, $Mg^+$ and $Na^+$, is demonstrated above 150 km, with the ratio between the lighter ions ($Mg^+$ and $Na^+$) and heavier ions ($Fe^+$) increasing with height by more than a factor of 2 above 150 km. More satellite observations of the $Mg^+/Fe^+$ ratio are needed to test this prediction. (6) The role of $NO^+$ and $O_2^+$ transport in the distribution of metal ions in the model is examined by comparing the two simulation results. It is found that they have little effect on the monthly means of metal ions but affect the peak heights of metallic ions in the descending phase of the "fountain effect".

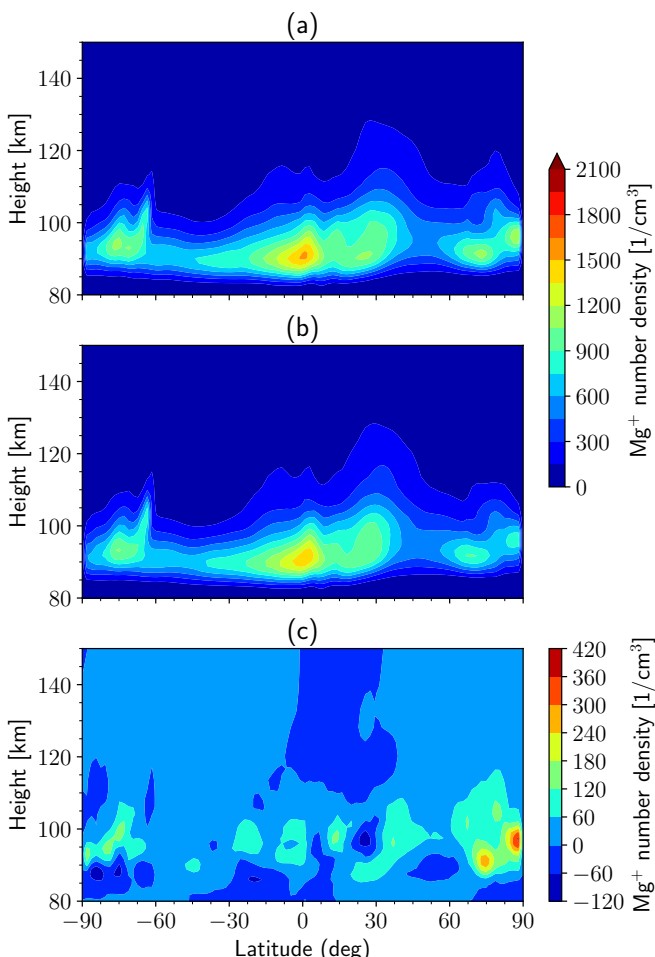

**Figure 7.** The monthly and zonal mean of the $Mg^+$ number density in June from (a) the TA simulation and (b) the CE simulation. (c) The difference between the TA and CE simulations.

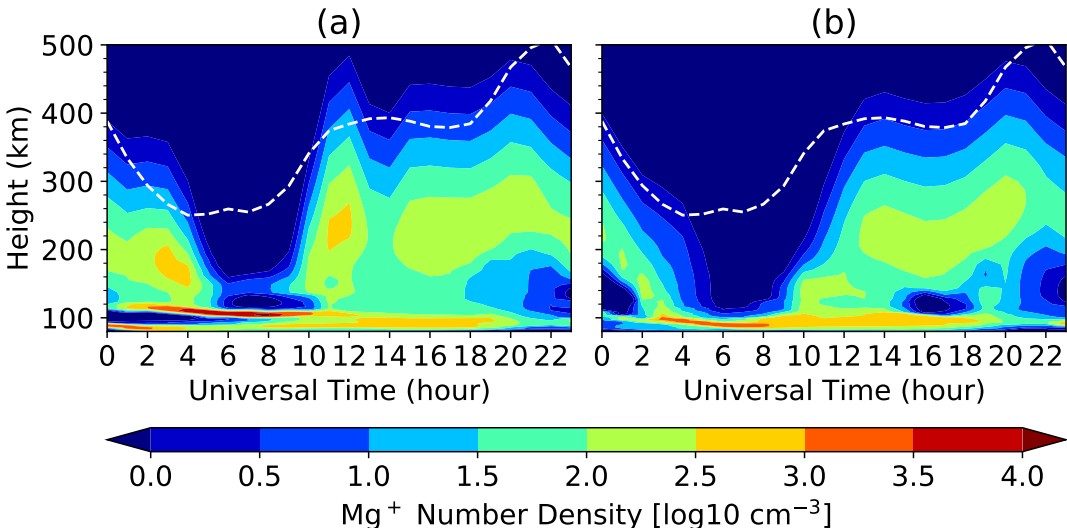

**Figure 8.** Mg$^+$ density over (12°N, 0°, near the dip equator) as a function of universal time and altitude at June Solstice. The dashed lines denote hmF2. (a) for the TA simulation and (b) for the CE simulation.

Previous research has established that thermospheric neutral metal layers are modulated by dynamics (e.g. gravity waves, atmospheric tides) (e.g., Chu et al., 2011; Xue et al., 2013; Liu et al., 2016; Cai et al., 2017; Qiu et al., 2016; Chu et al., 2020). In the future, this new version of WACCM-X can be used to better investigate the effect of lower atmospheric dynamical processes on the formation of thermospheric neutral metal layers, by using the "specified-dynamics" version of the model (SD-WACCM-X).

*Code and data availability.* The WACCMX model output has been archived at " National Space Science Data Center, National Science & Technology Infrastructure of China" (https://dx.doi.org/10.12176/01.60.00002-V01), and the simulation data sets used in this study are freely available from NSSDC Space Science Article Data Repository (https://dx.doi.org/10.12176/01.99.00212). As a part of CESM2, WACCM-X is available at http://www.cesm.ucar.edu/models/cesm2/.

*Author contributions.* JW, WF, HL, and JMCP designed the simulations and wrote the manuscript. XX and DRM contributed to the discussion and explanation of model simulations. All authors discussed the results and commented on the manuscript at all stages.

*Competing interests.* The authors declare that they have no conflict of interest.

*Acknowledgements.* This work was supported by the B-type Strategic Priority Program of the Chinese Academy of Sciences (Grant No. XDB41000000), the National Natural Science Foundation of China (42074181, 42125402, 41831071, 41804147, and 41704148), and the Open Research Project of Large Research Infrastructures of CAS "Study on the interaction between low/midatitude atmosphere and ionosphere based on the Chinese Meridian Project." . J. W. was funded by the Joint Open Fund of Mengcheng National Geophysical Observatory (No. MENGO-202008). W. F. and J. M. C. P. were funded by the European Research Council CODITA project (291332). HLL acknowl-

edges partial support by NSF OPP 1443726. National Center for Atmospheric Research is a major facility sponsored by the National Science Foundation under Cooperative Agreement No. 1852977. The numerical calculations in this paper were in part undertaken on the supercomputing system in the Supercomputing Center of University of Science and Technology of China, and ARC3, part of the High Performance Computing facilities at the University of Leeds. The authors would like to thank the National Space Science Data Center, National Science & Technology Infrastructure of China.

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
