# Peer review of "Self-consistent Global Transport of Metallic Ions with WACCM-X"

_Atmospheric Chemistry and Physics, 2021_

## Author Comment (AC1)

We thank the reviewers for making very useful suggestions to improve the paper. Our point-by-point responses to the reviewers' comments and corresponding changes with line numbers are detailed below in blue text, and the changes are shown in the version of the manuscript with track changes.

**Reviewer #3 comments (RC1):**

Understanding the global metallic ions transport in the thermosphere is important for the field. There is no doubt that WACCM-X is one of the excellent global-scale models to study metallic ions' transport. Overall, the work on this subject is worthy for publication. However, there are a few points that are not clear enough in the manuscript. I would like to suggest to the authors do a minor revision to clarify all the comments. Here are the detailed comments,

Major Comments:

First and foremost, the ambipolar diffusion velocity in equation 2 of the ion velocity equation adopts the equations (5.54) and (5.70) in Schunk and Nagy (2000), which include the effects of the neutral collision, the number density gradient, and temperature gradient, the gravity force, and the viscous stress along with the magnetic field line. Since the equation 5.54 is derived from the momentum equations 5.51 and 5.52, the neutral-ion momentum transform has been already taken into account. If the authors directly use equation 5.54 in Schunk and Nagy (2000), that could lead to double counting neutral wind effects in equation 2 in the manuscript. The ambipolar diffusion equation shows up as different formulas in various literature. It has to be careful to use them directly, and a strict mathematical derivation is required.

**Response:** We very much agree with the reviewer that a strict mathematical derivation is required. For this reason, when we used equation (5.54) in Schunk and Nagy (2000), we removed the field-aligned neutral wind in equations (5.54). In the model, metallic ions should be treated as minor ionic species at higher altitudes ($\geqslant 150$ km), and their movement will be influenced by the major species. In our formulation, we separate the bulk motion transport (neutral wind and various drifts) from those due to kinetic effects (pressure balance, including gravity). In order to avoid a misunderstanding, we have changed "given by" to "derived from" in the revised manuscript.

In section 3.1, the peak altitude of Mg+ is ~10 km higher in the summer hemisphere, and the authors suggest that it is caused by the summer to winter neutral wind that transports the metallic ions along with the magnetic field line. Although the vertical drift velocity is provided, the evidence is not enough to support this inference. Firstly, the authors need to clarify the contributions of electric field and neutral wind to the upward drift velocity of ~5 m/s in line 98. Does it mean the ~5 m/s upward drift velocity is only driven by neutral wind? Secondly, since the magnetic field line is roughly symmetric on both sides of the dip equator, we would expect the downward drift velocity on the other side. Some features in Figure 1 may indicate the downward drift, however, more clear evidence should be provided. Thirdly, due to lack of collision, the weak ion-neutral collision may lead to less effect of neutral wind in higher altitudes, but the upward drift velocity somehow increases with respect to altitude in Figure 1. If it is caused by the increasing winds, related evidence should be provided.

**Response:** Thank you for this comment. "The upward drift velocity of ~5 m/s" in line 98 of the previous manuscript is only driven by the neutral-ion collision-induced ion motion along the magnetic field lines. Indeed, the downward drift velocity is on the other side (see the second line of panels in Figure R1). Previously, Figure 1 mainly showed the upward drift velocity, but in the revised manuscript we illustrate both the upward and downward drift velocity. Please see the response to Reviewer #2 where this figure appears.

Compared with Figure 1, Figure 2 shows the Mg+ is barely above 100 km at 10 LT for all months, and much lower than those in Figure 1. It is skeptical because the wind and electric field show no upward effects at all, especially, the diurnal variations of Mg+ in section 3.2 show the Mg+ ions are easily transported to higher altitudes by the fountain effect around the local noon. Is there any explanation, and further investigation on this feature?

**Response:** The pronounced difference in Fig. 2 is due to the diurnal variation of ion electro-dynamical transport. Figure 4e in the revised manuscript shows the diurnal variation of $Mg^+$ number density near the dip equator as a function of local time. The $Mg^+$ drifts down towards the main layer at 10 LT. The figure shows that the fountain effect transports metal ions upward from 12 LT. The dominant chemical reactions at different heights are different, in particular the rapid neutralization of $Mg^+$ at lower heights (below 100 km, neutralization occurs with a time constant < 1 hour (Whalley et al., 2012)). In addition, as mentioned above, tides and other dynamic processes also play an important role in ion transport, and these vary with latitude.

In section 3.3, sentence "..., the ions at subtropical latitude (orange line) are transported upward to a small extent at midday, but transported downward to a lower height (~1 cm−3 at ~100 km) at March ..., which is in reasonable agreement with the downward drift of the fountain effect along the magnetic field lines at midnight." is questionable. The dynamo effect originates from the strong ion-neutral coupling in the lower E region, and the electric field produced by the dynamo effect could have substantial effects on the vertical transport of ions in higher altitudes. In the mid and low latitude region, the electric field may not be that important below 120 km, compared with neutral winds. In other words, neutral winds could be the dominant factor that determines in the vertical motion below 120 km in the mid and low latitude region. I would like to suggest the author examine the wind and electric field at March Equinox and June Solstice, and calculate the vertical drift velocities driven by neutral wind and electric field, respectively. In addition, it is hard to judge what factors determine the Fe+ profile shape for the orange and green lines. It's better to extend the discussion on the role of electric field and neutral winds.

**Response:** This is a good suggestion. The vertical drift velocity due to terms 1 - 4 of Equation 2 in the manuscript, the sum of the drift velocities, and the neutral vertical velocity at 12 LT at March Equinox and June Solstice are shown in Figure R5. The first and second terms are the Lorenz force-induced V×B drift and the neutral-ion collision-induced ion motion along the magnetic field lines, respectively. The third term is the Coulomb-force-induced ion drift along the E direction, and the fourth term is the E×B drift. Figure R5 shows that the drift velocity has a significant latitudinal distribution, in which V×B drift and Coulomb-force-induced drift are most significant from 110 km to 140 km (1st term and 3rd term), and transport along the magnetic field line and E×B drift are most significant above ~130km (2nd term and 4th term). At 12 LT, in general, the vertical ion

velocity driven by electric field (which is responsible for the fountain effect) plays a dominant role at low latitudes, while the vertical ion velocity due to neutral winds is dominant at mid-latitudes.

[Figure]

**Figure R5:** The vertical ion velocity due to terms 1-4 of Equation 2, the sum of the ion velocities, and the neutral vertical velocity at midday (12 LT), at March Equinox and June Solstice.

In section 3.4, what are the initial global number densities and distributions of Fe+, Mg+, and Na+? The studies in Huba et al (2019) show that there is not much difference between the transports Fe+ and Mg+, please refer to Figures 1 and 2 in Huba et al (2019). To clarify this issue, it's better to compare the vertical drift velocities of Fe+ and Mg+.

**Response:** The metal atoms and ions were initialized from a long-term simulation of WACCM-metals (e.g., Feng et al., 2017) and interpolated to the WACCM-X vertical grids with zero metal neutral/ions values above $6\times10^{-6}$ hPa. Their initial distributions are roughly Gaussian-shaped layers with peak heights between 90 and 100 km, similar to the Fe and $Fe^+$ from the control run in Figure 5 (dotted/dashed lines).

Figure R6 shows the altitude dependence of several transport coefficients ($\alpha$ defined in the same way as in equation 3 of Huba et al. (2019)). The coefficients of $Mg^+$ and $Fe^+$ are pretty similar, so

that our conclusion is similar to Huba et al (2019): "There is more than a factor of 2 difference in the masses of $Fe^+$ and $Mg^+$, the difference in the $\alpha$ factors is only ~10%"

The mass separation in WACCM-X is caused by the mass-dependent transport terms, particularly, molecular diffusion which does not appear to have been included in Huba et al. (2019) and should not be ignored above the turbopause. The WACCM-X molecular diffusion coefficient for minor species is described in the first WACCM-X paper (Liu et al. (2010), equation 5).

$$D_j = \frac{\sqrt{T}}{\rho}\overline{m}\sqrt{\left(\frac{1}{\overline{m}}+\frac{1}{m_j}\right)\frac{1}{Av}}1.52 \times 10^{20} \text{ m}^2\text{s}^{-1} \qquad (5)$$

Molecular diffusion will displace the species heavier than the background atmosphere (i.e., 29 amu.) downward.

[Figure]

**Figure R6:** $\alpha$ defined same as the equation 3 in Huba et al. (2019). The figure is similar to Figure 1 of Huba et al. (2019).

Feng, Wuhu, Bernd Kaifler, Daniel R. Marsh, Josef Hoffner, Ulf-Peter Hoppe, Bifford P. Williams and John M. C. Plane, Impacts of a sudden stratospheric warming on the mesospheric metal layers, J. Atmos. Sol.-Terr. Phys., http://dx.doi.org/10.1016/j.jastp.2017.02.004, 2017.

Minor Comments:

Line 35, "thermospheric metal atom" -> "the thermospheric metal atom"

**Response:** Corrected.

Line 43, "affects of ion" -> "effects of ion"

**Response:** Corrected.

Line 80, "a major sources" -> "a major source"

**Response:** Corrected.

Line 86, "$Na^+$, $Fe^+$ and $Mg^+$"->" $Na^+$, $Fe^+$ , and $Mg^+$"

**Response:** Corrected.

Line 102, "which shows a minimum" -> "which show a minimum"

**Response:** Corrected.

Line 115, "The modelled" -> "The modeled"

**Response:** ACP uses UK English so we have not made this change.

Line 120, "Diurnal variation of" -> "Diurnal variations of"

**Response:** Corrected.

Caption Figure 3. "… The white dashed lines indicates the position of the the dip equator." -> "… The white dashed lines indicate the position of the dip equator."

**Response:** Corrected.

In Figure 4, the caption says the time is UT, but the titles of all subplots are LT. It is confusing. Please make a consistent statement.

**Response:** Although UT and LT are the same at latitude 0°, we agree this is confusing. We have changed them to UT in Figure 4.

Line 170, "is thought to be related" -> "are thought to be related"

**Response:** Corrected.

Line 173, "Figure 7 and 8 compares" -> "Figure 7 and 8 compare"

**Response:** Corrected.

Line 184, "charge transfer of" -> "the charge transfer of"

**Response:** Corrected.

The axes of all figures are not friendly. Please add minor ticks in y-axis. Please add ticks from the low limit to the up limit of every axis.

**Response:** Thanks for this suggestion. We've updated the axes of all the figures to make them more user-friendly.

The x-axis of Figure 4 and Figure 8 better shows from 0 to 24 with 2 or 4 hours step.

**Response:** Thanks. Now changed.

---

## Author Comment (AC2)

We thank the reviewers for making very useful suggestions to improve the paper. Our point-by-point responses to the reviewers' comments and corresponding changes with line numbers are detailed below in blue text, and the changes are shown in the version of the manuscript with track changes.

**Reviewer #1 comments (RC2):**

General comments:

This is an interesting study on model simulations on Mg, Na and Fe (and their ions) in the mesosphere/thermosphere using an extended version of the WACCM-X model. In my opinion the study is suitable for ACP after some revisions have been made. Two more general aspects that will appear again in the specific comments below are:

(1) the differences between the Mg+ results presented here and the ones based on WACCM-Mg in Langowski et al. should be explicitly discussed and perhaps explained – if possible – by differences in the model set-up.

**Response:** This is a good suggestion. In the revised manuscript, we now describe the similarities and differences between our model results and the WACCM-Mg model and SCIAMACHY observations in Langowski et al. (2015). We have also added the model set-ups to facilitate the comparison with WACCM-Mg. The following briefly introduces the similarities and differences between the two models:

Similarities: As we stated in Section 2.1 (Line 52), "Validated metal chemistry modules for magnesium (Langowski et al., 2015), sodium (Marsh et al., 2013a), and iron (Feng et al., 2013) with updated rate coefficients from Plane et al. (2015), Bones et al. (2016) and Viehl et al. (2016), are added." Therefore, for Mg, our model is essentially the same as the Mg chemistry module in WACCM-Mg, and WACCM-X 2.0 and WACCM-Mg are extended from WACCM4 and have similar physical and chemical processes.

Differences: (1) The WACCM-X includes a self-consistent solution of global electrodynamics and extends the model top to more than 500 km, while the top of the WACCM-Mg model is about 140 km. (2) To make the model results more consistent with the observations by SCIAMACHY, the MIF adopted in the WACCM-Mg model by Langowski et al., 2015 was adjusted by scaling the Mg MIF to match the observed $Mg^+$ column density. (3) WACCM-Mg uses the specified dynamics version (SD-WACCM), which is nudged by the GEOS5 meteorological data set (including temperature, specific humidity, horizontal winds) below 60 km, while the model in the present paper is a free-running simulation that does not attempt to model specified lower atmosphere forcing.

The following table briefly summarizes the similarities and differences between WACCM-Mg and WACCM-X-Mg, taking Mg as an example.

|  | WACCM-Mg | WACCM-X-Mg |
| --- | --- | --- |
| Model Framework | CESM1 (Based on WACCM4) | CESM2 (Based on WACCM4) |
| Metal Chemistry | Magnesium chemistry modules (Langowski et al., 2015) | Magnesium chemistry modules (Langowski et al., 2015) |
| Model top | $6 \times 10^{-6}$ hPa (~140 km) | $4 \times 10^{-10}$ hPa (500-700 km) |
| Ion transport | No | full ion transport |
| E dynamo | No | Yes |
| Specified Dynamics | Nudged by the GEOS5 | Free-running |
| Meteoric Input Function (MIF) | Adjusted by scaling the Mg MIF to match the observed $Mg^+$ column density by SCIAMACHY. | Updated MIF considers more types of sources (Carrillo-Sánchez et al., 2016), but not adjusted to observations |

**Changes:** Please see Lines 54-55, Lines 100-106 and Lines 113-116.

(2) I agree that there are similarities between the simulations presented here and the SCIAMACHY observations in Langowski et al. (which is very good!), but there are also quite significant differences (e.g., in [Mg+] peak altitude or absolute number densities) which are not mentioned at all. Please discuss the similarities and differences in an objective manner and also include the differences.

**Response:** Thank you for this suggestion to discuss the similarities and differences in greater detail. In the revised manuscript we now describe the difference between the $Mg^+$ number density and peak height in our model and the SCIAMACHY observations, and also provide possible reasons for the differences.

**Changes:** Please see Lines 100-106.

Specific comments:

Line 4: "The model with full ion transport significantly improves the simulation of global distribution 5 and seasonal variations of Mg+."

I don't think this is entirely correct. The WACCM-Mg simulations shown in Langowski et al. were in better agreement with SCIAMACHY observations in terms of number density than the WACCM-X simulations presented here.

**Response:** Thank you for pointing this out. We have modified the sentence in the revised manuscript to read: "The model with full ion transport significantly improves the simulation of global distribution and seasonal variations of $Mg^+$, although the peak density is slightly lower (by on average 5 km) compared with the SCIAMACHY measurements."

For the global and seasonal distribution of Mg$^+$, our model compares better with observations. Of course, as the reviewer points out, WACCM-Mg is more consistent in terms of number density. We provide possible reasons in the following point-by-point responses.

Line 15: "and Friedman et al. (2013) investigated a descending thermospheric K layer up to 155 km at Arecibo"

This statement is a little vague. Did the layer descend from altitudes above 155 km downwards to 155 km? The combination of "descending" and "up to" makes the sentence difficult to understand.

**Response:** The sentence is rephrased to the following: "and Friedman et al. (2013) investigated a thermospheric K layer that descended from altitudes above 155 km at Arecibo, Puerto Rico."

Line 33: "However, no previous studies appear to have examined the full transport of metal ions in a self-consistent global chemical-dynamical model."

I'm not sure what "self-consistent" really means here. "Self-consistent" is used several times throughout the manuscript and at least in some cases with a different meaning. Can you explain, what it means here?

**Response:** what we mean here by "self-consistent" is that WACCM-X contains atmospheric dynamics (of the whole atmosphere up to ~500 km), chemistry (including the injection of the three metals from meteoric ablation), and ion transport, all solved numerically in a single model. In previous modelling work on ion transport of metallic species, most of the background physical quantities are input from empirical models rather than solved self-consistently. For example, neutral winds have been specified by the HWM empirical model and the thermospheric composition and temperature by NRLMSISE.

Line 43: "affects" -> "effects" ?

**Response:** Corrected.

Line 47: "WACCM-X is developed in the present study"

I suggest replacing "developed" by "extended", because WACCM-X did already exist before.

**Response:** Changed.

Same sentence: "combined with interactive chemistry"

Please mention briefly in what sense the chemistry is interactive. Often the chemistry is not "fully" interactive.

**Response:** "Interactive chemistry" in the model is relative to "specified chemistry", which refers to the full coupling of chemistry and atmospheric physics – in particular, heating through exothermic chemical reactions, and cooling through radiative emission from chemical species. As stated in Line 51 "The key chemistry and dynamical features are based on CAM4 and WACCM4 and are described in detail in Marsh et al. (2013)." WACCM4, as described in Marsh et al. (2013), includes fully interactive chemistry.

Marsh, D. R., Mills, M. J., Kinnison, D. E., Lamarque, J., Calvo, N., & Polvani, L. M. (2013). Climate Change from 1850 to 2005 Simulated in CESM1(WACCM), Journal of Climate, 26(19), 7372-7391.

Line 53: "and winds is treated in the same way as most active chemical species"

Which species are treated in a different way? Why?

**Response:** In WACCM, most chemical species are treated as advected tracers. However, a few species, including $NO^+$, $O_2^+$ and $N_2^+$, are short-lived so that transport is less important, and they are treated as being in photochemical steady-state in the model.

Perhaps "as most" -> "as for most" ?

**Response:** changed to "…as the transport of most …"

Line 56: "including a self-consistent electrodynamics module"

Can you mention briefly, what "self-consistent" means here?

**Response:** The term "self-consistent" is used here in the same way as before – the atmospheric physics, dynamics and chemistry is all treated in a single model"

According to Liu et al. (2018), the electric field (and along with it the E×B drift) at low and mid-latitudes in WACCM-X 2.0 is solved according to the electric dynamo equations, using neutral winds and electric conductivities resolved by the model. In earlier version of the model (WACCM-X 1.0), the electric field (and E×B drifts) had to be specified by an empirical model.

Liu, H.-L., Bardeen, C. G., Foster, B. T., Lauritzen, P., Liu, J., Lu, G., … Wang, W. (2018). Development and validation of the Whole Atmosphere Community Climate Model with

thermosphere and ionosphere extension (WACCM-X 2.0). Journal of Advances in Modeling Earth Systems, 10, 381– 402. https://doi.org/10.1002/2017MS001232

Line 61: "constant F107=124"

I suggest to replace "F107" by "F10.7". Also: The 10.7 cm flux is not dimensionless. Please add sfu (solar flux units) to the numerical value.

**Response:** Thank you for pointing this out. Now it is rephrased to the following: "(constant F10.7=130 sfu (solar flux units) and Kp=1)".

Line 95: "At middle latitudes .. , the peak altitude of Mg+ is 10 km higher in the summer hemisphere,"

Higher compared to which region? The same latitudes in the winter hemisphere? Or the rest of the latitudes? Please specify.

**Response:** We have modified the sentence in the revised manuscript to: "The peak altitude of $Mg^+$ in the summer hemisphere at middle latitudes ($\sim40\pm10°$) is $\sim10$ km higher than at other latitudes, in accord with observations (Langowski et al., 2015)."

Line 104: "To address this, we also present the simulation results at the same local time (10:00 LT) in Figure 2, and they are in better agreement with SCIAMACHY observations."

Yes, in some respect these results are in better agreement with the SCIAMACHY observations, but the number densities are even lower and the discrepancy to the number densities observed by SCIAMCHY is even bigger. In addition, the Mg+ peak altitude is still about 10 km lower than in the SCIAMCHY data set. These differences should also be explicitly discussed in my opinion, not only the aspects that fit well.

**Response:** Thank you for this suggestion. For the global and seasonal distribution of $Mg^+$, our model results are in better agreement with the SCIAMACHY observations. In terms of number density and peak altitude, there is a larger discrepancy with the SCIAMCHY observations. We have now added a description of the difference between the modelled $Mg^+$ number density and peak height and the SCIAMACHY observations, and also give possible reasons for the differences.

**Changes:** Please see Lines 100-106 and Lines 113-116.

Line 107: "which is absent in the previous models."

It is unclear – at least to me - what "previous models" refers to. Previous studies by other groups, WACCM-MG?

**Response:** This has been changed to refer specifically to Langowski et al. (2015), which reported in Section 7.2: "The Mg$^+$ peak altitude is close to 95 km and shows no strong variation with latitude and time."

Line 110: "the Mg+ column density exhibits relatively high distributions at"

I suggest replacing "high distributions" by "high values". "High distributions" doesn't really make sense here.

**Response:** Changed as suggested.

Section 3.1: Please explain the differences between the extended WACCM-X used in this study and WACCM-Mg used Langowski et al.. The WACMM-Mg results agreed much better with SCIA in terms of [Mg+]. Any idea why? In my opinion these differences are an important aspect that should be addressed in the paper.

**Response:** As we stated in Section 2.1 (Line 52), "Validated metal chemistry modules for magnesium (Langowski et al., 2015), sodium (Marsh et al., 2013a), and iron (Feng et al., 2013) with updated rate coefficients from Plane et al. (2015), Bones et al. (2016) and Viehl et al. (2016), are added." Therefore, for Mg, our model is basically consistent with the chemical scheme in WACCM-Mg, and both WACCM-X 2.0 and WACCM-Mg are based on WACCM4 and have similar physical and chemical processes.

The biggest difference are that WACCM-X includes global electrodynamical transport of the metallic ions (equation (2) in the manuscript), and extends the model top to more than 500 km (the top of WACCM-Mg model is about 140 km).

The major reason why the Mg$^+$ densities from WACCM-Mg are in better agreement with SCIAMACHY is because the MIF adopted by Langowski et al. (2015) was scaled to match the observed Mg$^+$ column density. Another possible reason is that WACCM-Mg used specified dynamics (SD-WACCM), where the meteorological fields (including temperature, specific humidity and winds) below 60 km are nudged by the GEOS5 meteorological data set. In contrast, WACCM-X-Mg is a free-running simulation.

Another difference is that we have used an updated MIF in the present study which considers more sources of interplanetary dust particles (Carrillo-Sánchez et al., 2020), compared with the MIF used by Langowski et al. (2015). Because the model simulates Na, Fe, and Mg simultaneously, and the present paper mainly focuses on the global transport processes of metal ions, we have not scaled the MIF to make the simulations conform to the observations.

Carrillo-Sánchez, J. D.; Gómez-Martín, J. C.; Bones, D. L.; Nesvorný, D.; Pokorný, P. Benna, M.; Flynn, G. F.; Plane, J. M. C. (2020) Cosmic dust fluxes in the atmospheres of Earth, Mars, and Venus, *Icarus*, *335*, art. no. 113395

Also section 3.1: There are of course similarities between your Mg+ results and the SCIAMACHY observations, but there are also significant differences. The Mg+ peak altitude in your simulations is about 10 km lower than in the SCIAMACHY data. Also, the modelled number densities are about a factor of 2 lower than in the SCIAMACHY data set. These differences should also be explicitly mentioned in the paper and possible reasons discussed. I'm not asking for new simulations, just an objective description of the agreement between model and measurement results.

**Response:** See above.

Fig. 2: Here, the low bias compared to SCIAMACHY is even more pronounced than in Fig. 1. Any idea why?

**Response:** The pronounced discrepancy in Fig. 2 is probably due to the diurnal variation of ion electro-dynamical transport. Figure 4e in the revised manuscript shows the diurnal variation of $Mg^+$ number density near the dip equator as a function of local time. This shows that $Mg^+$ drifts down towards the main layer at 10 LT, where removal of $Mg^+$ is more rapid.

Fig. 3: Units below colour bar is wrong/incorrect. Please correct.

**Response:** Changed the unit of column density to $10^9$ atoms $cm^{-2}$.

Also, the MgII VCDs shown here are significantly lower than the ones determined from the SCIAMACHY observations (compare to Fig. 16 (right) in Langowski et al.).

**Response:** We have updated Figure 3 using different color ranges. Compared with the $Mg^+$ VCDs simulated by WACCM-Mg (Fig. 16 (right) in Langowski et al. (2015)), the VCDs in Fig. 3e in our paper are comparable in value. Compared with the SCIAMACHY observations (Fig. 9 in Langowski et al. (2015)), although the VCDs in Fig. 3e are slightly lower in value (by 20%), there is good agreement with the observed global distribution and seasonal variation.

Line 173: "Figure 7 and 8 compares" -> " Figures 7 and 8 compare" ?

**Response:** Corrected.

Line 186: "In contrast, the reduced densities of the molecular ions (and electrons) at night means that their increased lifetimes become comparable to transport lifetimes."

Is the logic behind this sentence entirely correct? Please check.

**Response:** The sentence is rephrased to the following: "In contrast, the reduced densities of the molecular ions (and electrons) at night means that their increased lifetimes become comparable to the transport time-step."

Fig. 6: The Mg+/Na+ ratios exhibit an interesting interhemispheric difference for solstice conditions. Please comment on it and perhaps provide a qualitative explanation, if possible.

**Response:** Thank you for pointing this out. We have now provided the following explanation at Lines 178-190: "Figure 6 shows that whereas the $Mg^+/Fe^+$ ratios (left panel) show relatively little latitudinal variation compared with the variation with altitude (mostly caused by the mass difference of the ions), the $Mg^+/Na^+$ ratios (right panel) show marked interhemispheric differences at the solstice periods. The reason for this is the difference in the ion-molecule chemistry of $Na^+$, compared with $Mg^+$ (and $Fe^+$). The $Na^+$ ion has a closed electronic shell (it is isoelectronic with the inert gas Ne), and so does not react with $O_3$, in contrast to $Mg^+$ and $Fe^+$ (Plane et al., 2015). Formation of $MgO^+$ by the fast reaction with $O_3$ is the main route to neutralization of $Mg^+$ above 90 km (Whalley et al., 2011). During summer at mid- to high-latitudes (> 30°), $O_3$ above 90 km is heavily depleted through a combination of longer diurnal photolysis and reaction with the elevated levels of H produced from $H_2O$ which upwells over the summer pole (Plane et al., 2015). In contrast, the $O_3$ density in the lower thermosphere is more than an order of magnitude higher in the winter polar vortex. The result is that lower thermospheric $Mg^+$ ions at latitudes higher than ~30° are relatively long-lived in summer, and can be transported vertically throughout the thermosphere. This leads to the higher $Mg^+/Na^+$ ratios in the summer hemisphere, as shown in Figure 6. The converse operates at latitude higher than 30° in the winter hemisphere, where the relatively high $O_3$ tends to neutralize lower thermospheric $Mg^+$. Because $Na^+$ and $Mg^+$ have very similar masses, their ratios above 100 km are fairly constant with height."

Whalley, C. L., J. C. Gomez Martin, T. G. Wright, and J. M. C. Plane (2011), A kinetic study of Mg+ and Mg-containing ions reacting with $O_3$, $O_2$, $N_2$, $CO_2$, $N_2O$ and $H_2O$: implications for magnesium ion chemistry in the upper atmosphere, Physical Chemistry Chemical Physics, 13, 6352-6364.

Line 190: "the full life cycle of multiple meteoric"

I'm not sure what this really means: "the full life cycle". Is the formation of meteoric smoke particles included as well?

**Response:** The "full life cycle" in this paper is a phrase used to emphasise that our model simulates all relevant processes of metal species in the atmosphere (of which we are aware), including the injection rate of each element into the atmosphere as a function of time and latitude (the meteoric input function (MIF)), the metal chemistry, the transport of metal atoms and ions, and removal of the metallic species as particles.

To treat the polymerization of the metal reservoir species into meteoric smoke particles (MSPs), each metal chemistry module is assigned a "dimerization" reaction, where formation of the dimer represents permanent removal. For more information, please refer to Section 5.2 of Plane et al., (2015).

Plane, J. M. C., Feng, W., and Dawkins, E. C.: The mesosphere and metals: chemistry and changes, Chemical Reviews, 115, 4497–541, https://doi.org/10.1021/cr500501m, 2015.

Line 195: "(1) A clear seasonal cycle is found in the monthly averaged global distributions of Mg+, in good agreement with the SCIAMACHY measurements"

I agree the seasonal cycle is in good overall agreement with the SCIAMACHY measurements. However, several other aspects show significant differences (e.g. peak altitude, absolute values etc.). The differences should also be mentioned.

**Response:** Thank you for pointing this out. The sentence is rephrased to the following: "A clear seasonal cycle is found in the monthly averaged global distributions of $Mg^+$, in good agreement with the SCIAMACHY measurements (Langowski et al., 2015), although the peak height and peak density are about 5 km and 35% lower than the observations, respectively."

---

## Author Comment (AC3)

We thank the reviewers for making very useful suggestions to improve the paper. Our point-by-point responses to the reviewers' comments and corresponding changes with line numbers are detailed below in blue text, and the changes are shown in the version of the manuscript with track changes.

**Reviewer #2 comments (RC3):**

This work is an implementation of transports of metallic ions (Mg+, Fe+, and Na+) in WACCM-X for the purpose of understanding the effects of atmospheric motions on their distributions. The work lacks sufficient description of the implementation as well as scientific insights.

The implication of the transport effect, mainly described by equation (2), is questionable. There are terms missing as compared with Chu and Yu (2017), 2nd and 6th term in their eq(6). This is not explained. The effect of neutral wind (6th term) is especially important in the E region.

**Response:** The ion transport velocity described in equation 2 is an additional metal ion velocity term which has been added to the standard WACCM-X model. The 6th term of eq (6) in Chu and Yu (2017) is already included in the dynamical core of the WACCM-X model, and so is not included in equation 2 for simplification. We have therefore added the following sentence at Line 80 to make this clear: "Note that the term corresponding to transport is omitted from Equation 2, because advection by the neutral wind is already included in the dynamical core of WACCM-X."

The second term of eq (6) in Chu and Yu (2017) is the field-aligned term of the drift in electric field. We include this term in our model, but classify it as one of the ambipolar diffusion terms. This is consistent with Chu and Yu (2017), who stated: "The E•B term is not negligible, because it is important for field-aligned diffusion. This term is proportional to the electron pressure gradient along B. The electron and ion pressure gradient terms act in the same direction, **producing ambipolar diffusion**."

"Self-consistency" appears in numerous places but there is no explanation of what "self-consistency" really means, and what was not consistent in any previous model works. The difference of this work from that by Langowski et al. (2015) is not clear, except that the model used is WACCM-X instead of WACCM.

**Response:** see the response to reviewer #1.

Almost all conclusions were drawn based on comparisons of the distributions from model simulations with observations or other model runs. This is a poor way to gain much insight into any physical/chemical processes. Most of the conclusions are speculative, without any quantitative assessments. The benefit of modeling is to enable the examination of individual effects associated with different processes. If only the final results are examined, the value of modeling is not utilized much. Several examples are listed below:

**Response:** The main purpose of the present paper is to describe a new global thermospheric metal model with the ability to simulate the global transport of metallic ions. This is a challenging task in itself because of the very different time-scales of the various transport processes. Therefore, we

mainly present here comparisons of the model with/without electrodynamic transport to illustrate the importance of including electrodynamic transport of the long-lived metallic ions, and then evaluate the model performance by comparing with the existing observations and simulations. We are grateful to the reviewer for the suggestions and ideas they have provided, which will doubtless form part of our further work with the new model. We now address the specific points in turn.

1. The authors argue that the "upward transport of Fe+ does not significantly contribute to changes of Fe," (line 152).   To prove this, the contributions of Fe+ transport to the time tendency of Fe need to be calculated separately and compared with other effects.

**Response:** We have modified the relevant sentence in the revised manuscript to: "In general, changes in Fe (at densities > 1 cm$^{-3}$) do not appear to be synchronized with the upward transport of Fe$^+$, so that the vertical distribution of Fe in the transport run is similar to that in the control run."

Fe/Fe$^+$ ion-neutral coupling is important in the formation of thermospheric iron layers, as shown particularly by the work of Xinzhao Chu's group at UC Boulder. At present, there are many unsolved mysteries about this, which partly inspired us to develop the new model. However, as we stated above, this article mainly introduces our new global thermospheric metal model and its ability to simulate the global transport of metallic ions. As stated in the Conclusions Section of the manuscript "In the future, this new version of WACCM-X can be used to better investigate the effect of lower atmospheric dynamical processes on the formation of thermospheric neutral metal layers, by using the "specified-dynamics" version of the model (SD-WACCM-X)."

2. The authors argue that "uplift of metal ions" is "driven by "meridional wind," (line 196-197).   This could be supported if the authors can demonstrate that the magnitude of uplift due to the meridional wind is indeed much larger than other wind components.

**Response:** The vertical ion velocity due to terms 1 - 4 of Equation 2 in the manuscript, and due to the neutral wind (term 1 + term 2) are shown below in Figure R1 for March and June. The first and second terms are the Lorenz force-induced V×B drift and the neutral-ion collision-induced ion motion along the magnetic field lines, respectively. The lowest panels in Figure R1 show the drift velocity resulting from the combined effect of the first and second terms. Figure R1 shows that the drift velocity caused by the neutral wind has a significant latitudinal distribution, in which V×B drift mainly operates from 110 km to 140 km, and transport along the magnetic field line mainly works above ~130km. Therefore, from the continuity equation, the uplift in the mid-latitudes of the summer hemisphere is mainly driven by the vertical ion velocity due to the neutral wind.

We have rephrased the sentence in the revised manuscript to: "This appears to be caused by the vertical ion velocity due to the neutral wind (first and second terms in Eq. 2), with the monthly mean drift velocities shown in Figure 1".

[Figure]

**Figure R1:** The monthly and zonal mean of vertical ion velocity due to terms 1-4 of Equation 2, and due to neutral wind in March and June.

3. The authors believe that this uplift can "explain the summer maximum occurrence of thermospheric sodium layers." If that were the case, then the contribution of the uplift term to the production of neutral sodium would show being much larger than others. Furthermore, the authors did not even show that the model actually produced a summer maximum of Na in the thermosphere. In addition, it is not true that thermospheric Na appears mostly in summer. As Cai et al. (2017) stated, the thermospheric sodium appeared at this Southern Hemisphere site mostly in spring and fall and rarely in summer and winter.

**Response:** The model simulations show that the peak altitude of $Mg^+$ in the summer hemisphere at ~40° is ~10 km higher than at other latitudes, which is very similar to the SCIAMACHY observations (Langowski et al. 2015). The Beijing and Utah State University lidar stations are located at ~40° N. Both stations report that the occurrence of high-altitude sporadic Na layers (Nas) and lower thermospheric Na layers peak during summer, which is consistent with the latitudinal distribution of the metal ions we simulated. So we express it as "this is in good agreement with the summer peak occurrence of lower thermospheric neutral Na layers observed by mid-latitude lidars (Wang et al., 2012; Dou et al., 2013; Yuan et al., 2014; Xun et al., 2020).".

Cai et al. (2017) used model simulations to investigate the summertime Na variations at mid-latitudes, in particular the occurrence peaks of the high-altitude Nas during summer at Utah State University lidar station. Cai et al. (2019) used simulations to investigate thermospheric sodium layers (TSLs) at Cerro Pachón, Chile ((30.25°S, 70.74°W). Indeed, they found that TSLs appeared at Andes Lidar Observatory (ALO) mostly during equinox. However, ALO is located at 30.25°S (20°S in magnetic latitude), which is different from the phenomenon described by the present paper. At the same time, this shows the diversity of TSL formation mechanism. Further work on the geophysical mechanisms of their formation is clearly required.

As stated in the Conclusions Section in the manuscript "Previous research has established that thermospheric neutral metal layers are modulated by dynamics (e.g. gravity waves, atmospheric tides). In the future, this new version of WACCM-X can be used to better investigate the effect of lower atmospheric dynamical processes on the formation of thermospheric neutral metal layers, by using the "specified-dynamics" version of the model (SD-WACCM-X)."

Most of the thermospheric Na observed showed a downward phase progression similar to that of diurnal or semidiurnal tides.  Since tides are well resolved in WACCM-X, this feature should be reproducible if the model results are to be believed.  On the other hand, the thermospheric Fe at high latitude showed variations on much shorter timer scales (Chu and Yu 2017).  Comparison with Fe observations with a model that cannot resolve the short-time scale dynamics (gravity waves) is not meaningful.

**Response:** We have not proposed that the present model is appropriate for studying the effects of short-wavelength gravity waves on the thermospheric Fe layers. For that, a higher resolution (both horizontally and vertically) version of WACCM-X will be needed. For example, Liu et al. (2014) developed the first mesoscale-resolving whole atmosphere general circulation model (high-resolution WACCM), which can simulate mesoscale gravity waves. A combination of our metal ion transport model and this regionally refined general circulation model should be available in the near future for studying metallic layers.

Liu, H.-L., McInerney, J. M., Santos, S., Lauritzen, P. H., Taylor, M. A., and Pedatella, N. M. (2014), Gravity waves simulated by high-resolution Whole Atmosphere Community Climate Model, Geophys. Res. Lett., 41, 9106– 9112, doi:10.1002/2014GL062468.

The reviewer is correct that a tidal influence is observed in the model. Figure R2 shows the vertical ion velocity due to terms 1-4 of Equation 2 in the manuscript, and due to neutral wind (term 1 + term 2). The first and second terms are related to the background wind, that is, the neutral wind tides, and the tidal influence on the vertical ion velocity is clear in the figure. Figure R3 shows the diurnal variation of $Mg^+$, demonstrating that atmospheric tides strongly influence the distribution and transport of the metal ions. The corresponding Mg diurnal variation is shown in Figure R4. The topside of the Mg layer shows significant diurnal variation.

[Figure]

**Figure R2:** The vertical ion velocity due to terms 1-4 of Equation 2, and due to neutral wind.

[Figure]

**Figure R3:** Time and altitude distributions of Mg$^+$ number density.

[Figure]

**Figure R4:** Time and altitude distributions of Mg number density above 105 km.

Other minor points

According to Liu et al. (2018), WACCM-X "neglects the influence of ion-neutral collisions on ion motion perpendicular to B," which is significant "in the E-region where the O+ lifetime is short and transport is unimportant."   Since E-region is an area of focus in this study, how does this neglected effect influence the simulation result?

**Response:** As described in Section 2.2.2 of Liu et al. (2018), it is reasonable to keep $O^+$ in chemical steady-state equilibrium in WACCM-X because the $O^+$ lifetime in the E region is short (because of its rapid reaction with $N_2$), and transport is less important. Note that transport of metal ions in our model includes ion motion perpendicular to B (refer to equation 2 in the manuscript).

169-170: TIFe and TINa are not defined.   In fact, there is no point in using such acronyms as they are mentioned only once and they are not widely accepted.

**Response:** Suggestion incorporated.

Figure 3e needs to adopt a different color range to show more structure.

**Response:** Thanks for this suggestion. We have updated Figure 3 using different color ranges.

I suggest that the authors cite the following paper, which is the first report of thermospheric Na in the southern hemisphere.   Furthermore, the temperature and wind reported in this paper is a rare dataset that can be used to validate model simulations trying to reproduce the thermospheric Na.

Liu, A. Z., Guo, Y., Vargas, F., and Swenson, G. R. (2016), First measurement of horizontal wind and temperature in the lower thermosphere (105–140 km) with a Na Lidar at Andes Lidar Observatory, Geophys. Res. Lett., 43, 2374– 2380, doi:10.1002/2016GL068461.

**Response:** This reference has now been added.